# THOI: An efficient and accessible library for computing higher-order interactions enhanced by batch-processing

Laouen Belloli[1,2,3☯]*, Pedro A. M. Mediano[4,5], Rodrigo Cofré[6], Diego Fernandez Slezak[1,2], Rubén Herzog[7,8☯]*

**1** Laboratorio de Inteligencia Artificial Aplicada, Instituto de Ciencias de la Computación, Universidad de Buenos Aires, Buenos Aires, Argentina, **2** Instituto de Investigación en Ciencias de la Computación (ICC), CONICET-Universidad de Buenos Aires, Buenos Aires, Argentina, **3** Institut du Cerveau, Paris Brain Institute, ICM, Inserm, CNRS, Sorbonne Université, Paris, France, **4** Department of Computing, Imperial College London, London, United Kingdom, **5** Division of Psychology and Language Sciences, University College London, London, United Kingdom, **6** Universite Côte d'Azur, INRIA CRONOS Team, Sophia Antipolis, France, **7** Instituto de Física Interdisciplinar y Sistemas Complejos (IFISC, UIB-CSIC), Palma de Mallorca, Spain, **8** Department of Psychology, University of the Balearic Islands, Palma de Mallorca, Spain

☯ These authors contributed equally to this work.
* laouen.belloli@gmail.com (LB); rherzog@ifisc.uib-csic.es (RH)

## Abstract

Complex systems are characterized by nonlinear dynamics, multi-level interactions, and emergent collective behaviors. Traditional analyses that focus solely on pairwise interactions often oversimplify these systems, neglecting the higher-order interactions critical for understanding their full collective dynamics. Recent advances in multivariate information theory provide a principled framework for quantifying these higher-order interactions, capturing key properties such as redundancy, synergy, shared randomness, and collective constraints. However, two major challenges persist: accurately estimating joint entropies and addressing the combinatorial explosion of interacting terms. To overcome these challenges, we introduce THOI (Torch-based High-Order Interactions), a novel, accessible, and efficient Python library for computing high-order interactions in continuous-valued systems. THOI leverages the well-established Gaussian copula method for joint entropy estimation, combined with state-of-the-art batch and parallel processing techniques to optimize performance across CPU, GPU, and TPU environments. Our results demonstrate that THOI significantly outperforms existing tools in terms of speed and scalability. Specifically, THOI reduces the time required to exhaustively analyze all interactions in small systems (≤ 30 variables). For larger systems, where exhaustive analysis is computationally impractical, THOI integrates optimization strategies that make higher-order interaction analysis feasible. We validate THOI's accuracy using synthetic datasets with parametrically controlled interactions and further illustrate its utility by analyzing fMRI data from human subjects in wakeful resting states and under deep anesthesia. Finally, we analyzed over 900 real-world and synthetic datasets, establishing

**Data availability statement:** All data underlying the results presented in this study are publicly available from external repositories. The anesthesia analysis dataset is available from the OpenNeuro repository at https://openneuro.org/datasets/ds003171/versions/2.0.1. The dataset comprising synthetic and real-world systems analyses is available from the Zenodo repository at https://zenodo.org/records/7118947. No new data were generated for this study.

**Funding:** The author(s) received no specific funding for this work.

**Competing interests:** The authors have declared that no competing interests exist.

a comprehensive framework for applying higher-order interaction (HOI) analysis in complex systems. THOI opens new perspectives for testing both established and novel hypotheses about the multi-level, nonlinear, and multidimensional nature of complex systems.

## Introduction

Complex systems are characterized by nonlinear dynamics, intricate interactions spanning multiple organizational levels, and emergent collective behaviors that cannot be fully explained by the properties of individual components [1–4]. Understanding these systems is challenging due to the presence of complex, often poorly understood interdependencies, which traditional model-based approaches struggle to address.

Conventional methodologies typically focus on pairwise interactions, relying on simplifying assumptions that fail to capture the richness of these interdependencies. While pairwise models can provide useful insights, they neglect higher-order dependencies—interactions involving three or more elements simultaneously—that are critical for explaining the emergent phenomena observed in complex systems. By reducing the analysis to pairwise relationships, such approaches risk producing an incomplete or even misleading representation of the system's true dynamics and underlying processes [5–7].

To address the limitations of traditional pairwise approaches, information theory provides a powerful and general framework for quantifying the informational structure of complex systems. By extending Shannon's mutual information to account for higher-order interactions (HOI), this framework enables the identification and quantification of statistical dependencies that go beyond linear and pairwise correlations [8–11]. These higher-order dependencies are essential for understanding the emergent behaviors and intricate interdependencies inherent in complex systems.

At the heart of information theory lies the concept of entropy, which measures the unpredictability or randomness of a system and represents the average amount of information gained from observing it [12,13]. In neuroscience, for instance, entropy has been applied to evaluate neural variability [14,15] and to capture the complexity of information processing in the brain [16–18]. This makes information theory particularly well-suited for studying the intricate dynamics of systems where higher-order dependencies play a critical role.

HOI refer to interdependencies involving three or more variables, capturing collective behaviors that cannot be reduced to pairwise relationships. By accounting for these interactions, HOI provide a deeper understanding of complex dependencies within systems. Multivariate information theory offers a rigorous framework to study HOI through several extensions of mutual information, including the total correlation ($TC$), dual total correlation ($DTC$), O-information ($\Omega$), and S-information [8] (see Supplementary Methods). Each of these metrics reveals distinct aspects of higher-order dependencies: $TC$ quantifies collective constraints, $DTC$ represents shared

randomness, $\Omega$ captures the balance between synergy and redundancy, and S-information reflects the overall level of interdependence.

Synergy and redundancy are key components of HOI. Synergy refers to information that emerges only when the system is analyzed as a whole and cannot be inferred from individual parts, while redundancy represents repeated information distributed across the system. These measures are derived from various linear combinations of low- and high-order entropies, unified under the entropy conjugation framework [19].

In this work, we focus on $\Omega$, as it uniquely assesses the quality of interactions, i.e., synergy or redundancy dominance, rather than simply measuring the overall level of interdependence. Since its introduction, $\Omega$ has provided novel insights across diverse fields, including whole-brain dynamics [9,20], altered states of consciousness [21,22], spiking neural networks [23,24], macroeconomic trends [25], and music analysis [26,27].

Despite its broad applicability, the use of $\Omega$ faces two significant challenges: (i) like other information-theoretic measures, it requires the estimation of joint probability distributions, which often necessitates large datasets that may be difficult to obtain; and (ii) the combinatorial explosion of possible HOI, which scales exponentially with the number of variables (e.g., a system with 30 elements yields approximately $2^{30} \sim 10^9$ HOI).

To mitigate some of the challenges in analyzing complex systems, several open-source libraries have been developed for estimating entropy, mutual information, and related measures [28–32]. While some of these tools include estimators for $\Omega$ and other HOI, they are often not optimized for large-scale analyses or for seamless accessibility in standard computational environments.

To address these limitations, we introduce THOI (Torch-based High-Order Interactions), a novel Python library specifically designed for efficient computation of HOI in large systems. THOI leverages the Gaussian copula (GC) method [30,33] for joint entropy estimation, which enables direct computation from the covariance matrix of GC-transformed data. This approach bypasses the need for direct probability distribution estimation, significantly reducing computational complexity (see Supplementary Methods). To further enhance efficiency, THOI integrates with PyTorch [34], using optimized batch matrix operations to exploit the parallel processing capabilities of modern hardware, including CPUs, GPUs, and TPUs.

We evaluated THOI's performance by comparing its computational efficiency to existing open-source libraries for estimating $\Omega$. For large systems where exhaustive computation of all possible interactions is infeasible, we implemented and validated optimization algorithms that balance accuracy and scalability using synthetic datasets with known ground-truth $\Omega$ values.

Finally, we demonstrate the practical utility of THOI through two applications: analyzing functional magnetic resonance imaging (fMRI) data to reveal reductions in synergistic interactions during deep anesthesia compared to wakefulness, and benchmarking its efficiency by analyzing over 900 multivariate datasets (both real-world and synthetic) in under 30 minutes on a standard laptop.

## Materials and methods

### Multivariate information theory

We first introduce the fundamental concepts of information theory, focusing on entropy and mutual information, before extending these ideas to HOI, with particular emphasis on the $\Omega$. As previously noted, entropy serves as a cornerstone of information theory, quantifying uncertainty or information content.

In what follows, we denote by $X^n$ a continuous multivariate random variable with $n$ components (or simply $X$ when referring to a single variable), and by $X_j$ the $j-th$ component of $X^n$. A specific realization of $X$ is represented as $x$, with $p(x)$ denoting its associated probability. Given that we are working with continuous data, we utilize Shannon's differential entropy, denoted as $H(X)$, defined as follows:

$$H(X) = -\int_x p(x)\log(p(x))dx$$

(1)

This formula requires knowledge of $p(x)$, i.e., an estimation of the probability density function (PDF). Unlike discrete entropy, differential entropy is unbounded and can take both negative and positive values, with units in nats when the natural logarithm is used.

For simplicity, we will refer to differential entropy simply as entropy throughout this work. The Eq 1 is valid for multivariate systems, where $H(X^n)$ denotes the joint differential entropy. This, in turn, depends on the joint probability density function (JPDF), and the integrals are taken over the entire support of the JPDF.

For a pair of continuous random variables $X$ and $Y$ (or a bivariate system $X^2$), the mutual information $I(X;Y)$ follows:

$$I(X; Y) = H(X) + H(Y) - H(X, Y) = \sum_{j=1}^{2} H(X_j) - H(X^2)$$

(2)

It can also be expressed in terms of conditional entropies:

$$I(X; Y) = H(X, Y) - H(Y|X) - H(X|Y) = H(X^2) - \sum_{j=1}^{2} H(X_j|X^2_{-j})$$

(3)

Here, $X^2_{-j}$ corresponds to the full system without the variable $X_j$. As previously mentioned, it informs about the uncertainty that is reduced on one variable when we know the other and is guaranteed to be non-negative. In the following, we present its generalizations for multivariate systems with $n>2$, where 2 and 3 are no longer equivalent.

**Generalizations of the mutual information**

The generalizations of 2 and 3 for higher order interactions have been called the total correlation ($TC$) [35], and the dual total correlation ($DTC$) [36]. They follow:

$$TC(X^n) = \sum_{j=1}^{n} H(X_j) - H(X^n)$$

(4)

$$DTC(X^n) = H(X^n) - \sum_{j=1}^{n} H(X_j|X^n_{-j})$$

(5)

Both the $TC$ and the $DTC$ are multivariate generalizations of the mutual information 2 and 3, respectively. However, they are not equivalent. They are non-negative quantities. $TC$ has been interpreted as collective constraints, while $DTC$ represents shared randomness.

The $\Omega$-information as proposed in [8] quantifies high-order interdependencies in complex systems by measuring the balance between redundancy and synergy. The $\Omega$-information is defined as the difference between the total correlation and the dual total correlation as follows:

$$\Omega(X^n) = TC(X^n) - DTC(X^n) = (n-2)H(X^n) + \sum_{j=1}^{n}[H(X_j) - H(X^n_{-j})]$$

(6)

Therefore, $\Omega$ can be expressed solely in terms of entropies. A system is said to be synergy-dominated if $\Omega < 0$, and redundancy-dominated if $\Omega > 0$. In other words, if $DTC(X^n)$ exceeds $TC(X^n)$, the system is considered more synergistic than redundant. This indicates that the interdependencies between the variables contribute information that cannot be inferred from examining the variables individually. Conversely, if $TC(X^n)$ exceeds the $DTC(X^n)$, the system is classified as more redundant than synergistic. This suggests that the interdependencies are largely due to shared or repeated information across the variables.

Finally, $S$-information is defined as the addition between the total correlation and dual total correlation as follows:

$$S(X^n) = TC(X^n) + DTC(X^n)$$

(7)

## Scalable batch-based architecture for higher-order information computation

The computational cost of calculating HOI is primarily driven by two factors: the need to compute the joint entropy for combinations of $k$ elements and the inherent combinatorial complexity of the problem (see Supporting information The combinatorial explosion). Calculating $\Omega$ is computationally intensive, with efficiency varying depending on the entropy estimator used.

In contrast, the entropy estimator for Gaussian variables, explained in Supporting information Entropy analytical expression for multivariate Gaussian variables, simplifies this process by requiring the calculation of the covariance matrix only once for the entire system. Subsequent sub matrices associated with each combination of variables ($n$-plet) can then be extracted efficiently, making this approach computationally less expensive. However, the number of possible combinations and the associated computational cost scale exponentially (see Supporting information The combinatorial explosion), making this an NP-hard problem. Although it cannot be fully solved, certain computational strategies, such as parallelization and batch processing, can significantly improve both time and memory performance.

To address the combinatorial explosion in computational complexity as the number of variables increases, we implemented a PyTorch-based batch-processing architecture [34]. This approach groups and processes data in parallel, significantly improving efficiency in the computation of HOI across large datasets.

The workflow (Fig 1) begins by transforming multivariate time series $X$ into covariance matrices $\Sigma$ using the Gaussian copula method (see Supporting information Estimation of entropy via Gaussian copulas). Binary masks are then applied to extract sub-covariance matrices $\Sigma^k$ for each $n$-plet of $k$ variables. These matrices are processed in batches to compute the determinants required for estimating entropies and HOI metrics such as $DTC$, $TC$, $\Omega$, and $S$-information (see Supporting information Generalizations of the mutual information).

This architecture enables the simultaneous processing of multiple datasets, allowing for real-time analyses such as identifying the $n$-plet that minimizes the average $\Omega$ across datasets. Additionally, users can easily control the batch size via a single parameter (*batch size*), ensuring scalability, efficient memory management, and compatibility with standard computational platforms.

A key feature of THOI is its ability to handle sub-covariance matrices of varying sizes within the same batch. Since different orders of interactions correspond to matrices of different dimensions, traditional batch processing becomes inefficient due to the fixed size of each batch, requiring iterative loops for matrices of different sizes. To overcome this limitation, we implemented independent variable padding, allowing computations for different interaction orders to be performed efficiently within the same batch (see Methods Batch processing of multiple orders of interactions using padded batches).

**Parallel evaluation of higher-order information measures.** In this section, we present how the Gaussian copula entropy estimator can be leveraged to compute the measures defined in Eqs 4, 5, 6, and 7 for multiple subsets of variables ($n$-plets) and for multiple datasets of the same size in parallel and in a batched fashion using PyTorch. This implementation allows us to exploit the power of matrix operations and batch processing on both CPUs and GPUs, as well

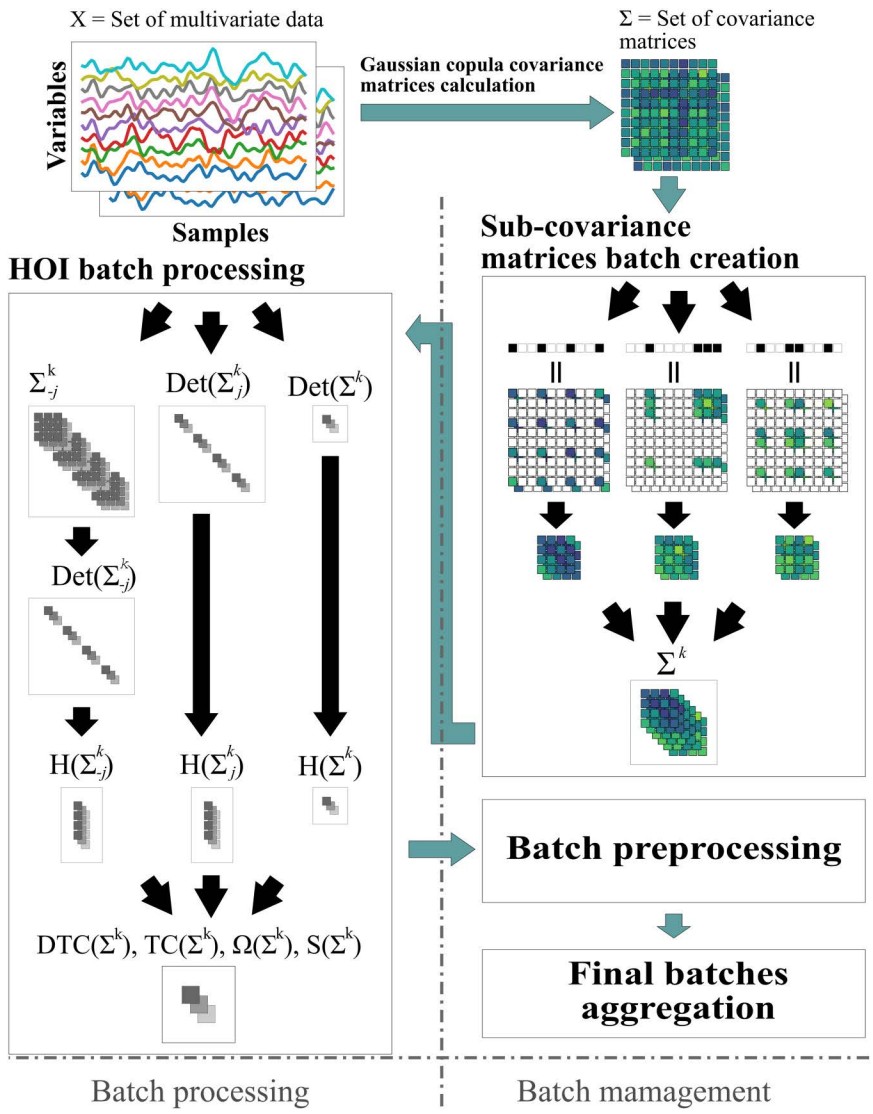

**Fig 1. Efficient computation of HOI using batch processing of covariance matrices.** A set of multivariate time series $X$ is transformed using the Gaussian copula approach, generating covariance matrices $\Sigma$ for each dataset. The covariance matrices are then sub-sampled using a batch of $k$-plet indices, defined by a binary mask applied to $\Sigma$, yielding the sub-covariance matrices $\Sigma^k$ for each k-plet. These sub-covariance matrices are then batched together, and their determinants are computed, which are subsequently used to calculate the entropies and associated HOI defined by the *DTC*, *TC*, $\Omega$, and *S*-information metrics, where the entropy is computed from the determinants of single variables $H(\Sigma_j^k)$, the whole system $H(\Sigma^k)$ and the whole system without a single variable $H(\Sigma_{-j}^k)$ (see Supporting information Generalizations of the mutual information for detailed descriptions). Finally, batches are pre-processed using a custom function (e.g., extracting the minimum $\Omega$), and the results are aggregated to produce the final output. Note that multiple datasets with identical system and sample sizes can be processed simultaneously and the batch management system allows flexible analysis on the fly.

as the parallelization capabilities of HPC architectures. The provided Python library depends only on PyTorch and NumPy and works seamlessly with or without CUDA (the library for performing computations on GPUs), providing an open-source, accessible, and ready-to-use Python library similar to NumPy.

Batch processing is feasible due to the following reasons:

1. **Independence of subsets**: The computations for different subsets of variables are independent, allowing for concurrent processing without interference.

2. **Linear algebra operations**: The core computations involve linear algebra operations (e.g., matrix slicing, determinant calculation) that are inherently parallelizable and can be vectorized using tensor operations.

3. **Advanced computing architectures**: Modern computing architectures available in ordinary laptops, such as GPUs and multi-core CPUs, are optimized for parallel computations on tensors, making batch processing computationally efficient.

To understand the core computations and why they can be processed in parallel, we will decompose the algorithm shown in Fig 1 into its components. From Eqs 4, 5, 6, and 7, we need to compute the following terms,

1. $H(X_j)$: The entropy of each random variable $X_j$.

2. $H(X^n)$: The entropy of the joint random variable $X^n$, i.e., the entire system.

3. $H(X_{-j}^n) = H(X_1, \ldots, X_{j-1}, X_{j+1}, \ldots, X_n)$: The entropy of the entire system excluding $X_j$.

Since all these terms are entropies of the form given in Eq 1, we can use the Gaussian copula approach (Eq 22 in Supplemental Methods) to estimate the full system's covariance matrix and extract the required *n*-plet sub-covariance matrices as a batch. Once we have the sub-covariance matrices, we can compute Gaussian entropies using Eq 14 (Supporting information) in a batched fashion, as all the operations are matrix computations available for batch processing in PyTorch. Finally, we compute the measures from Eqs 4, 5, 6, and 7 using only addition and subtraction operations.

To construct batches of sub-covariance matrices from the *n*-plets, we proceed as follows:

1. **Covariance matrices**: We consider a set of $D$ covariance matrices $\Sigma_d \in \mathbb{R}^{D \times N \times N}$ represents the covariance matrix of a dataset or a random vector of dimension $N$.

2. **$n$-plets**: We define a collection of $B$ *n*-plets $\{\mathbf{i}_b\}_{b=1}^B$, where each $\mathbf{i}_b \in \mathbb{Z}^K$ is an index set representing a subset of $K$ variables out of $N$.

3. **Batch formation**: We create a batch by pairing each *n*-plet with each covariance matrix, resulting in $B \times D$ combinations. This pairing is represented using high-dimensional tensors, enabling simultaneous processing.

4. **Sub-covariance extraction**: For each combination, we extract the sub-covariance matrix corresponding to the variables in the *n*-plet from the full covariance matrix. This extraction is achieved through tensor indexing operations, ensuring efficiency and parallelizability.

The following section describes in more detail the batch operations required to create the batched sub-covariance matrices.

**Tensor construction of batched sub-covariance matrices.** Let $\Sigma \in \mathbb{R}^{D \times N \times N}$ be the tensor of covariance matrices and $\mathbf{I} \in \mathbb{Z}^{B \times K}$ be the tensor of *n*-plets.

1. **Tensor expansion**:

- **Expand $n$-plet Indices**: We expand the *n*-plet indices to align with the dimensions of the covariance matrices:

$$\mathbf{I}_{\mathrm{exp}} \in \mathbb{Z}^{B \times D \times K},$$

(8)

where $\mathbf{I}_{\mathrm{exp}}[b, d, :] = \mathbf{i}_b$ for all $d$.

- **Expand covariance matrices**: We expand the covariance matrices to align with the batch of *n*-plets:

$$\Sigma_{\mathrm{exp}} \in \mathbb{R}^{B \times D \times N \times N},$$

(9)

where $\Sigma_{\exp}[b, d, :, :] = \Sigma_d$ for all $b$.

2. **Sub-covariance extraction**:

- **Row Selection**: For each combination $(b, d)$, we select the rows corresponding to the indices in $\mathbf{i}_b$:

$$\Sigma_{\text{rows}}[b, d, :, :] = \Sigma_{\exp}[b, d, \mathbf{i}_b, :].$$

(10)

- **Column selection**: We then select the columns corresponding to $\mathbf{i}_b$:

$$\Sigma^k[b, d, :, :] = \Sigma_{\text{rows}}[b, d, :, \mathbf{i}_b],$$

(11)

resulting in the sub-covariance matrices $\Sigma^k \in \mathbb{R}^{B \times D \times K \times K}$.

These operations are highly parallelizable, enabling the simultaneous extraction and computation of all required sub-covariance matrices. Moreover, by avoiding explicit loops and leveraging tensor operations, the computational overhead is significantly reduced. This approach scales efficiently with both the number of subsets and the size of the covariance matrices, making it well-suited for large-scale problems.

**Handling variable-order interactions via padded covariance batches.** One limitation of the previously discussed batch calculation of *DTC*, *TC*, *S*, and $\Omega$ is that batched *n*-plets must be of the same order to have the same size. This requirement arises because batched computations can only be broadcast over elements with identical shapes, and the sampled sub-covariance matrix of a given *n*-plet has a shape that depends on the size of the *n*-plet. This issue is common in deep learning models where inputs can have variable lengths, rendering batch processing challenging. A widely recognized foundational approach to handling variable-length inputs in deep learning—especially in the context of language modeling and sequence-to-sequence tasks can be found in early sequence modeling papers [37–39]. These studies popularized the practice of padding input sequences with special tokens so that all items in a batch share a uniform length, while the model learns to ignore the padding during training. For sub-covariance matrices, we cannot simply add a special token as padding because there are no training mechanisms to learn to ignore them. Instead, we add as padding an identity matrix. Since the added component is independent, we have the following:

$$\begin{aligned} H(\Sigma^k) &= H(\Sigma^k + I^{n-k}) - H(I^{n-k}) \\ &= H(\Sigma^k + I^{n-k}) - (n-k)H(N(0, 1)) \\ &= H(\Sigma^k + I^{n-k}) - (n-k) \cdot 1.4189, \end{aligned}$$

(12)

where $H(\cdot)$ denotes the entropy function, $\Sigma^k$ is the covariance matrix of the $k$ variables of the $n$-plet, 1.4189 is the entropy of a normal distribution with standard deviation 1, and $I^{n-k}$ is the identity matrix of size $n-k$.

Thus, we can create a batch of padded sub-covariance matrices by adding an independent normally distributed sub-component. We then perform computations in a batched manner as previously explained, and subsequently subtract the entropy of the added component, which is a known constant value multiplied by the length of the padding. Additionally, because the added component corresponds to a standard normal distribution covariance matrix, no bias correction needs to be applied to this sub-component. Fig 2 illustrates the proposed padding mechanism. It is worth noticing that in order to avoid unnecessary operations, the sampled sub-covariance matrix and the independent components are not sorted to be separated, but maintains the original positions of the variables in the full original covariance matrix.

While employing a padding strategy enables us to compute *TC*, *DTC*, $\Omega$ and *S* in a batched fashion, it is more memory-intensive because the batched covariance matrices are larger than the not padded version. Therefore, it is not

# Fixed lenght padded covariance matrix batch creation

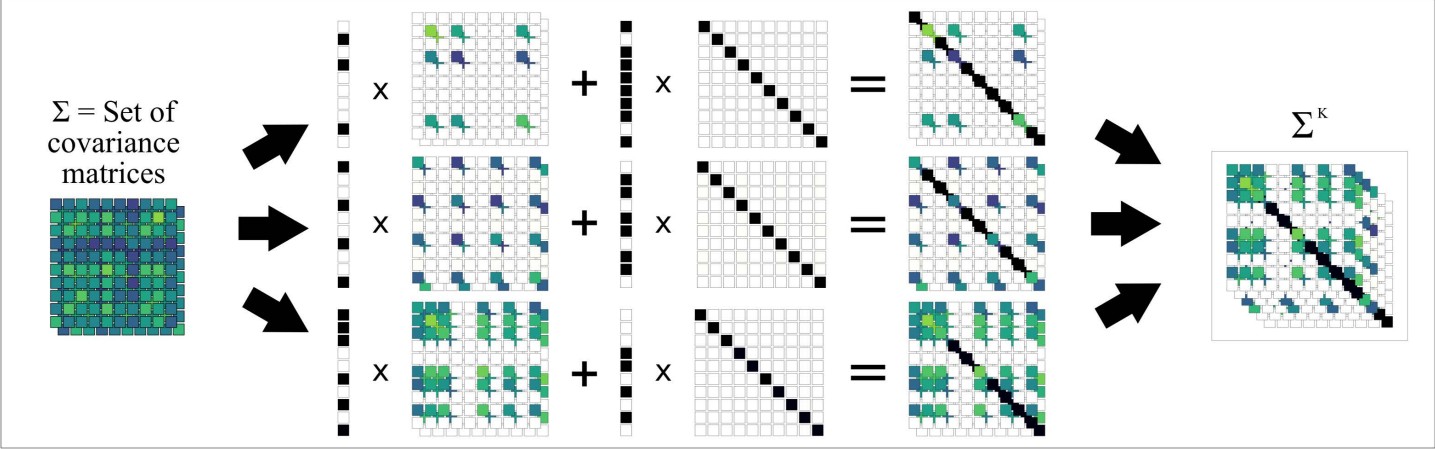

**Fig 2. Sub-covariance matrices sampled with padding to allow different covariance matrix sizes in a single batch. 1)** First, a mask is applied to the full covariance matrices using a masked encoding of the *n*-plets (each with a different number of masked variables) to obtain each sub-covariance matrix. At this point, the obtained covariance matrices are invalid as the masked rows and columns have zeros on the diagonal, yielding a constant distribution. **2)** Then, an identity matrix is masked with the inverted *n*-plet encodings. **3)** Both masked matrices are added to obtain the final covariance matrix where the rows and columns of the *n*-plet have the values from the full covariance matrix, and the remaining rows and columns have ones on the diagonal and zeros elsewhere, representing an independent standard normal component.

always advisable to use this strategy. If possible, sorting and processing batches by orders of interaction is preferable to avoid this extra overhead.

## Heuristics

Despite pushing the computational limits on the number of variables that can be processed in a reasonable time, systems with a larger number of variables still require strategies to estimate the $\Omega$ as the number of *n*-plets to compute grows exponentially (See Supporting information The combinatorial explosion). In this section, we describe two heuristic algorithms implemented in THOI: the greedy algorithm (GA) and simulated annealing (SA). These algorithms also process data in a batched fashion, enhancing their efficiency and allowing us to explore the space of *n*-plets more thoroughly to ensure that the obtained values are representative of the entire set.

   **Greedy algorithm.** GA are a fundamental class of algorithms in computer science and optimization, characterized by constructing a solution iteratively by selecting the best available option at each step based on the current state. They make locally optimal choices with the hope that these choices will lead to a globally optimal solution. Importantly, a GA does not reconsider its previous decisions, meaning it does not backtrack or revise its choices once made.
   **Key characteristics.**

- **Greedy Choice Property**: At each step, the algorithm makes the most advantageous choice based solely on the current state, without considering future consequences.

- **Optimal Substructure**: The problem can be broken down into smaller sub-problems, and an optimal solution to the overall problem contains optimal solutions to its sub-problems.

   While GA are powerful for certain types of problems and domains [40], their effectiveness depends on whether the problem exhibits the optimal substructure property.

We designed the GA to efficiently select subsets of variables (*n*-plets) of a given size (order of interaction) that maximize (or minimize) a given function over the *TC*, *DTC*, $\Omega$, and *S* measures and over a sequence of datasets. Traditional GA often evaluate candidates sequentially; however, we leverage PyTorch's batch processing capabilities to evaluate all possible candidates in parallel at each step. This approach allows the algorithm to systematically expand the current solutions with all possible new candidates, evaluate all the new solutions in a single batched step, and then efficiently select the optimal solutions.

At each iteration, the algorithm expands the current set of $\kappa$ solutions, each of size *t*, by adding every possible candidate variable not already included in the solution. Suppose we have *N* total variables and each current solution includes *t* variables; there are $N - t$ candidate variables remaining for each solution. This results in a total of $\kappa \times (N - t)$ new candidate solutions of size *t* + 1.

To manage this expansion efficiently, we represent the solutions and candidates using tensors and perform operations in a batched, matrix-oriented fashion. The process can be described as follows:

1. **Current Solutions Representation**: Let $\chi$ be a tensor of shape $(\kappa, t)$ representing the current $\kappa$ solutions, where each row corresponds to one solution containing *t* variable indices.

2. **Candidate Variables Identification**: For each current solution $\chi[j]$, we identify the set of valid candidate variables **K**[*j*] not yet included in $\chi[j]$. This results in a tensor **K** of shape $(\kappa, N - t)$, where $N - t$ is the number of remaining variables.

3. **Expansion to New Candidate Solutions**: We create a new tensor $\chi_{new}$ of shape $(\kappa, N - t, t + 1)$ to hold all possible expanded solutions. Each element $\chi_{new}[j, i]$ is constructed by concatenating the *j*-th current solution $\chi[j]$ with the *i*-th candidate variable **K**[*j*, *i*], resulting in a candidate solution of size *t* + 1. This can be all implemented using the same expanding and indexing strategy introduced in section Batched sub-covariance matrices.

4. **Batch Evaluation**: The tensor $\chi_{new}$ contains all possible new candidate solutions formed by adding one variable to each current solution. We can then evaluate all these $\kappa \times (N - t)$ candidate solutions simultaneously using batched operations in PyTorch.

5. **Selection of Optimal Solutions**: After evaluating the new candidate solutions, we select the top $\kappa$ solutions based on the optimization criterion (e.g., maximizing or minimizing the $\Omega$ measures). These selected solutions become the current solutions for the next iteration.

The final GA has the following steps:

• **Solution Initialization**: The algorithm starts with the top $\kappa$ *n*-plets of the initial size, obtained by performing an exhaustive search over that order of interactions.

• **Iterative Incremental Step**: At each iteration, we add one variable to the current $\kappa$ solutions by choosing from the set of variables not yet included, ensuring that the added variables are optimal at each step. This step is implemented using the previously explained strategy of batch expansion, calculation, and evaluation of all possible solutions for all $\kappa$ *n*-plets simultaneously.

• **Termination Criteria**: Once the optimal *n*-plets reach the desired order of interaction, the algorithm returns the obtained solutions.

**Simulated annealing algorithm.** SA is a stochastic optimization technique inspired by the physical process of annealing in metallurgy, where materials are heated and then slowly cooled to reach a stable state with minimal internal energy [41]. In computational terms, SA explores a solution space by initially allowing for random, high-energy moves to escape local optima, gradually decreasing the probability of such moves over time to settle into a global optimum.

For HOI analysis, SA complements the GA by enabling exploration beyond local optima, thus avoiding the limitations of non-optimal substructures. While a GA focuses on making the best immediate choice at each step, SA introduces randomness,

allowing it to occasionally accept suboptimal moves. This flexibility helps the algorithm navigate complex solution landscapes more thoroughly, overcoming the limitations of greedy methods in avoiding local optima and reaching a more globally optimal solution. Furthermore, because SA operates on a solution landscape where each solution is linked with related solutions by certain criteria (e.g., differing by one or more variables), it allows us to explore the space either inside an order of interaction or across order of interactions. To explore the landscape across orders or interactions, we transition from a solution at one order of interaction to a neighboring solution at another order by adding or removing a variable. Our implementation extends the classic SA algorithm by introducing batch processing, enabling the simultaneous optimization of multiple candidate solutions at once. This is achieved by representing the solutions as a batch tensor and provides better explorations of the space.

One challenge in evaluating multiple $n$-plets from different orders of interactions is that batch processing typically assumes all covariance matrices in the batch have the same size. To overcome this, we used the multi order batch processing explained in Section Batch processing of multiple orders of interactions using padded batches, which allows us to evaluate covariance matrices of different sizes in a single batch. The implemented SA algorithm proceeds as follows:

### Initialization.

- **Solution Representation**: Each solution is represented as a masked vector of length $N$, where $N$ is system size, a one in the $i$-th position means the $i$-th variable is in the $n$-plet, and a zero means it is not.

- **Initial Solutions**: A batch of $\kappa$ random solutions is generated, with each solution containing at least 3 elements to be all valid $n$-plets.

### Energy evaluation.

- **Objective Function**: An energy (or cost) function $E$ evaluates each solution. This function can be customized based on the problem and supports both maximization and minimization objectives.

- **Batch Evaluation**: The energy of all solutions in the batch is computed simultaneously, exploiting parallelism.

### Acceptance criteria.

1. **Energy Difference**: Calculate the change in energy $\Delta E = E_{\text{new}} - E_{\text{current}}$.

2. **Acceptance Probability**: A solution is accepted with probability:

$$
P = \begin{cases} 1, & \text{if } \Delta E > 0 \\ \exp\left(-\dfrac{\Delta E}{Temp}\right), & \text{if } \Delta E \leq 0 \end{cases}
$$

(13)

where $Temp$ is the current temperature.

3. **Update Rules**: If the new solution is accepted, it replaces the current solution; otherwise, the current solution remains unchanged.

### Cooling schedule.

- The temperature $Temp$ is initialized to a high value and decreases at each iteration according to a cooling rate $\alpha$ (e.g., $Temp_{\text{new}} = \alpha\,Temp_{\text{current}}$ with $0 < \alpha < 1$). Smaller $Temp$ yeld smaller probabilities of accepting solutions with negative energies.

### Termination criteria.

- The algorithm terminates after a fixed number of iterations or if no improvement is observed over a certain number of iterations (early stopping).

Both the GA and the SA approaches optimize an objective function defined over the $TC$, $DTC$, $\Omega$, and $S$ measures using the batch processing mechanism described in Section Efficient computation of HOI via Gaussian copulas, they inherently accommodate multiple datasets within a single optimization run. As a result, these methods readily support group-level analyses (e.g. population), facilitating the simultaneous consideration and comparison of multiple datasets in a unified framework.

### fMRI analysis

The anesthesia analysis described in section Analysis of human brain activity under anesthesia was conducted using a publicly available dataset of 17 healthy adults scanned on a 3T Siemens [42]. Each participant was recorded during resting-state at four propofol anesthesia levels: Awake (no propofol), Light, Deep, and Recovery (no propofol). We restricted the analysis to only the Awake and Deep. The dataset was already parcellated into 11 predefined networks, each composed of 5 distinct brain regions, resulting in a total of 55 regions per participant. We did not perform any additional preprocessing steps; all analyses were conducted directly on the data as it was published. One participant was excluded because their data did not include all 55 regions, leaving a final sample of 16 participants and two conditions (32 datasets in total).

We applied both GA (optimized at each order of interaction) and SA algorithm (optimized across multiple interaction orders) to identify the $n$-plets that either maximize or minimize the paired Cohen's $d$ effect size between the two conditions. A Wilcoxon signed rank paired test (each participant underwent the two conditions) with no correction for multiple comparisons was used when comparing between conditions.

### Complex systems dataset analysis

We employed a publicly available dataset from [43], selecting only those datasets containing at most twenty variables (obtaining a total of 920 final datasets). No additional preprocessing steps were performed. For each selected dataset, we exhaustively computed the full suite of HOI measures and subsequently derived the 21 metrics described in section Analysis of large database of synthetic and real-world systems.

## Results and discussion

We validated THOI by benchmarking its computational efficiency and scalability against existing open-source libraries, focusing on exhaustive HOI computations. For larger systems where exhaustive analysis is infeasible, we developed and tested optimization strategies, demonstrating their effectiveness on both synthetic and real-world datasets. Additionally, we highlight THOI's accessibility by analyzing over 900 distinct datasets in under 30 minutes on a standard laptop, showcasing its compatibility with commonly available computational setups.

### Enhanced performance

To evaluate the computational advancements of THOI, we compared its performance against three widely used open-source libraries for computing HOI:

**HOI_toolbox (Higher-Order Interactions Toolbox)** [31]: A Python library optimized for efficiently computing HOI in datasets using the Gaussian copula estimator [30]. It is designed for memory and processing efficiency but is limited to systems with $N \leq 20$ for exhaustive computations.

**HOI (High-Performance Estimation of Higher-Order Interactions)** [32]: A Python library designed to leverage high-performance computing (HPC) architectures. While it supports multiple estimators, it does not specifically address the combinatorial challenges associated with large-scale HOI computations.

**JIDT (Java Information Dynamics Toolkit)** [28]: A Java-based library primarily focused on analyzing information flow dynamics. It includes the Kraskov-Stögbauer-Grassberger (KSG) estimator [44] for entropy and related measures but does not tackle the combinatorial explosion problem inherent to HOI.

For benchmarking, we created a multivariate Gaussian system with 30 variables, each independently drawn from $\mathcal{N}(0, 1)$ (it is worth noticing time performance is not affected by the variable distribution and we only report the distribution for completeness). The goal was to compute $\Omega$ for all combinations of variables from order 3 to 30 without leveraging additional parallelization, ensuring a fair comparison. All computations were performed on a standard laptop equipped with an Intel Core i9 processor, 64 GB of RAM, running Linux Mint 21.

We observed a clear performance improvement with THOI compared to the other libraries (Fig 3A). Using THOI, we were able to compute all interaction orders from 3 to 30 in just 5.8 hours. In contrast, the other libraries were limited to order 8 within a reasonable computational time (<5 hours). The maximum computational time was expected to be close to 15, as the combinatorial explosion occurs at approximately $N/2$.

While computing all possible interactions with the other libraries was infeasible due to time or memory constraints, we extrapolated their computational times. For **HOI_toolbox**, this would have taken approximately 221 days. **HOI** would have required 2 days, with a memory overload of 240 GB just for $n$-plets indices creation and allocation. **JIDT** would have taken roughly 17 years to complete. Additionally, THOI can compute $TC$, $DTC$, $S$-information, and $\Omega$ in a single run, whereas other libraries require separate functions for each measure, which can increase computational time by a factor of 4. In terms of memory usage, THOI completed the full set of interaction orders using less than 3 GB of memory (with a batch size of 10,000). Finally, we assessed how computational time scales with sample size in a system of 20 variables. Again, THOI outperformed the other libraries in terms of computational efficiency (Fig 3B).

## Optimization for larger systems: Greedy and simulated annealing algorithms

Although THOI greatly accelerates the computation of HOI, exhaustively assessing all possible combinations in larger systems is computationally prohibitive. To address this, heuristic optimization algorithms, such as greedy algorithms (GA) [45] and simulated annealing (SA) [9,46], have been implemented in the THOI framework. These algorithms optimize an objective function –here $\Omega$ for demonstrative purposes– by targeting the most relevant combinations of variables, circumventing the need for exhaustive enumeration. The GA, deterministic in nature, optimizes locally at each order of interaction, using its outcome as the starting point for the subsequent order of interaction. In contrast, SA leverages stochasticity to escape local optima by allowing random changes early in the search, gradually refining solutions by reducing

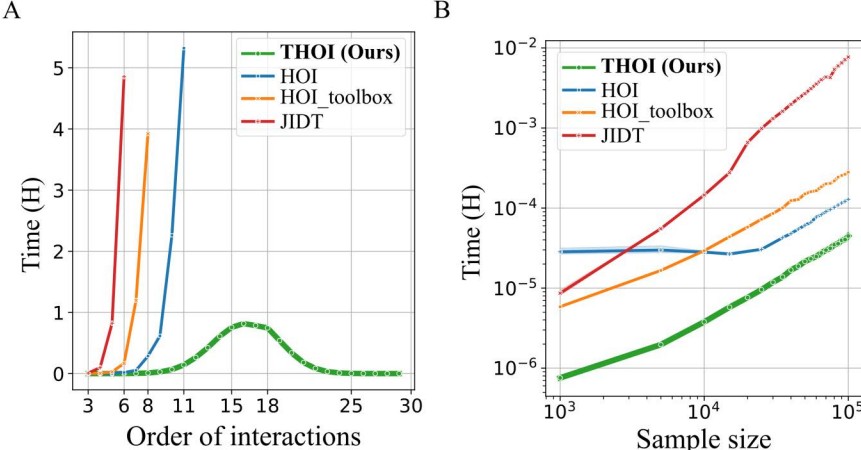

**Fig 3. Efficient computation of HOI using batch processing of covariance matrices. A)** Computational time versus order of interactions for a 30-variable system with 1000 samples. The THOI method successfully computes all possible HOI in less than 6 hours, whereas other libraries are unable to process interactions beyond order 11 within the same time frame. **B)** Log-log plot of computational time as a function of sample size for a 20-variable system. All libraries exhibit logarithmic scaling, but THOI outperforms the others in terms of computational speed.

randomness over time. Our implementation of the GA optimizes across multiple initial conditions, while the SA method employs both within-order optimization (focusing on a specific order of interaction) and across-order optimization (allowing transitions between solutions of different order of interaction).

To evaluate the accuracy of these algorithms, we designed a system of 100 variables consisting of five 20-variable sub-systems where the ground truth is known [19] (see Supporting information Probabilistic graphical models (PGM)): two redundant (R) systems (weak and strong), two synergistic (S) systems (weak and strong), and one independent system. The strength of each sub-system was determined by the coupling parameter $c$, weak = 0.5 and strong = 1, (see Supporting information Probabilistic graphical models (PGM)). Following the additive property of the $\Omega$ for independent components (i.e. $A \perp B \Rightarrow \Omega(A + B) = \Omega(A) + \Omega(B)$), this configuration allowed us to pre-define the maximal and minimal $\Omega$ values, corresponding to the sum of R and S sub-systems, respectively, and to ensure perfect synergy-redundancy balance ($\Omega=0$) for the whole 100-variable system.

We ran the GA and the SA for each order of interaction, finding the correct identity (i.e. the variables of the $n$-plets) and $\Omega$ values associated with each of the R and S subsystems with both algorithms (Fig 4). As we progressed from lower to higher orders, the maximum $\Omega$ increased monotonically, reaching order 20 (the size of the strong R system) and then slowed in its rise before saturating at order 40 (strong R + weak R). After this saturation point, the first decline occurred at order 99, when the entire weak S system was included, and then at order 100, when both S systems were completely included, achieving a perfect balance between synergy and redundancy (Fig 4C, 4D) expected by design. Conversely, minimum $\Omega$ decreased non-linearly up to 20 (strong S system), then at a slower rate until saturating at 40 (strong S + weak S). Beyond this point, the minimum $\Omega$ started to increase at order 60, where variables from the weak R system were included, continuing to decline as strong R variables were added, reaching zero (Fig 4E, 4F). This result not only confirms that the GA and SA heuristic algorithms function correctly for the given system, but also illustrates a key property of synergistic systems: the O-information increases non-linearly, with most of the information gain occurring only when all variables are present, and dropping sharply for incomplete subsets.

Because the GA seems to require significantly more repeats than the SA to achieve comparable performance, we conducted an additional test on a smaller system with 30 variables. We evaluated the algorithm's behavior using different numbers of repeats (2, 20, 200, and 2000) to assess how many were needed to correctly identify the optimal $\Omega$ solutions. Notably, the algorithm required the full 2000 repeats to find the optimal $\Omega$ solution (for a detailed explanation, see Supporting information Number of repeats in Greedy algorithm). Furthermore, as the SA algorithm is stochastic in nature, we tested the stability of its solutions when different initial conditions were used. We found that the solutions found by the two versions of the SA algorithm (i.e. order-free and order-specific) were highly stable in terms of the values and the variables selected in the optimal solution. See Supporting information Stability of simulated annealing across orders of interaction for an exhaustive description.

## Analysis of human brain activity under anesthesia

To demonstrate the utility of THOI in empirical research, we applied it to fMRI data from 16 subjects, each undergoing both wakeful rest and deep anesthesia. The data, previously published in [47], include brain activity across 55 brain regions, which are organized into 11 distinct brain networks, each containing five regions. This analysis was inspired by prior studies that have linked alterations in levels of consciousness to changes in brain complexity [17,18,47–49]. Our primary goal was to investigate whether anesthesia induces significant changes in the $\Omega$, reflecting shifts in the brain's HOI during these distinct states of consciousness.

We first applied the GA to independently obtain maximum and minimum $\Omega$ across different orders of interaction in both conditions (Fig 5A). The results revealed that deep anesthesia led to a reduction in both the maximum and minimum $\Omega$ values across all orders of interaction, indicating a significant decrease in brain complexity during this state. To identify groups of brain regions with the most pronounced differences between wakefulness and deep anesthesia,

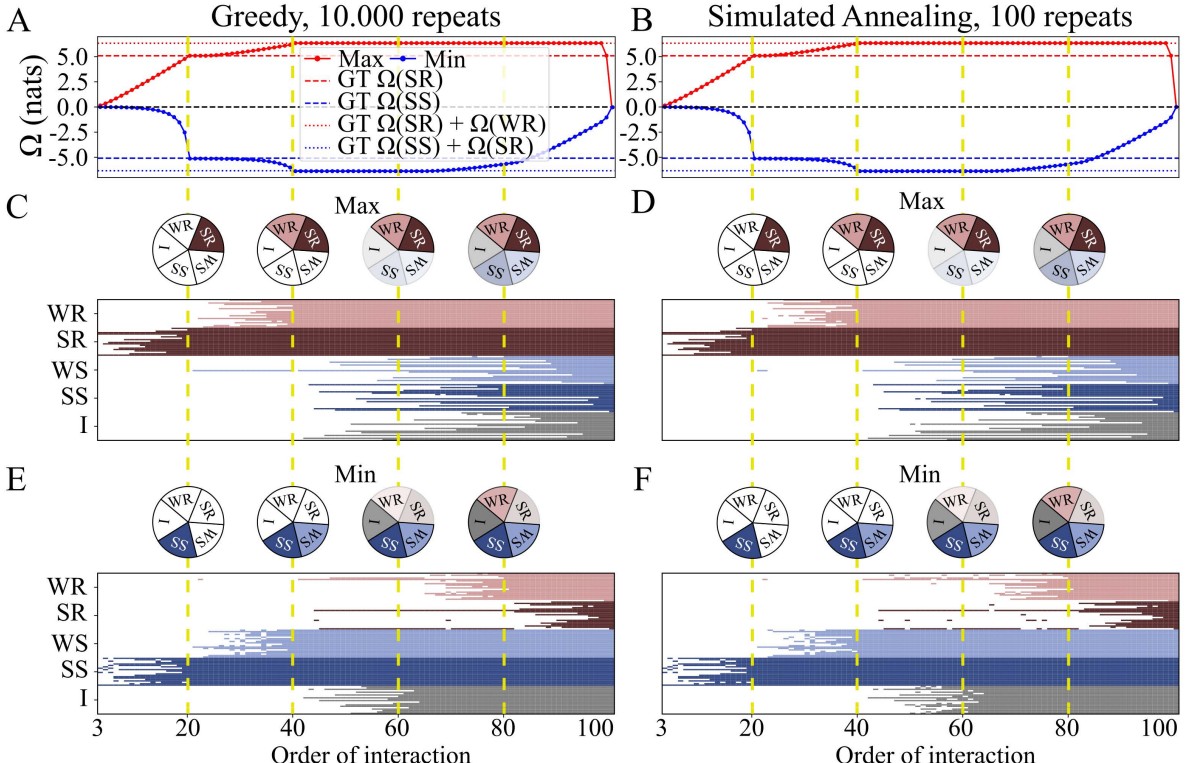

**Fig 4. Within-order optimization with greedy and simulated annealing algorithms A, B) Maximum (red) and minimum (blue) Ω obtained by greedy and SA algorithms for a 100-variable system composed of strong/weak R and S systems and an independent system, each with 20 variables.** Dashed horizontal lines indicate ground truth for the weak systems, and dotted lines represent the sum of weak and strong systems (red for R, blue for **S**). Both algorithms successfully identify the systems, but greedy required 100 times more repeats than SA. The yellow vertical dashed line denotes the order of interaction of the ground truth subsystems and their concatenation are detected, i.e., 20, 40, 60, 80 and 100. Note that for 20 and 40 a local and a global maxima (minimum) is detected, the former corresponding to the strong systems and the second to the concatenation of the strong and weak systems. **C, D)** Subsets of variables that maximize Ω at each order of interaction for greedy and SA. Each row corresponds to a single variable and colors denote different subsystems (weak R: light red; strong R: dark red; weak S: light blue; strong S: dark blue; independent: gray). Colored cells indicate that the variable contributed to maximize Ω at a given order of interaction, and white the opposite. The pie charts are positioned in line with the yellow vertical dashed line of panel **A** to summarize the subsystems that were detected. Both algorithms prioritize strong R, then weak R, followed by a mix of independent and S systems (denoted by weak color intensities in the pie charts). **E, F)** Subsets of variables that minimize Ω. Both algorithms prioritize strong S, then weak S, followed by a combination of independent and R systems, with a preference for the former.

we employed a GA to maximize and minimize effect sizes (separately) in Ω across both conditions (Fig 5B). While no significant differences were observed at the whole-brain level (Wilcoxon $p > 0.001$), Fig 5C), significant differences with large effect sizes were found (Cohen's $|d| > 3$, Wilcoxon test $p < 0.001$, not corrected) for both increases and decreases in Ω (Fig 5D, 5E).

The most notable increase in Ω involved eight regions from distinct resting-state networks [47]. This interaction, which was synergy-dominated during wakefulness, shifted to redundancy-dominated during anesthesia. Conversely, the largest decrease in Ω encompassed 14 regions, including four from the Cingulo-Opercular network, four from the Cingulo-Parietal network, and two from the Default Mode network, among others. Despite the reduction in Ω, these regions remained redundancy-dominated under both conditions. Using a SA algorithm across orders, we found results that mirrored those from the GA (Fig 5E), showing significant differences with large effect sizes for both Ω increases and decreases. Although the Ω increase was less than with the GA, it similarly resulted in synergy-to-redundancy shifts in some subjects. Reductions in Ω, again, did not result in synergy dominance, even with greater effect sizes than those found with the GA.

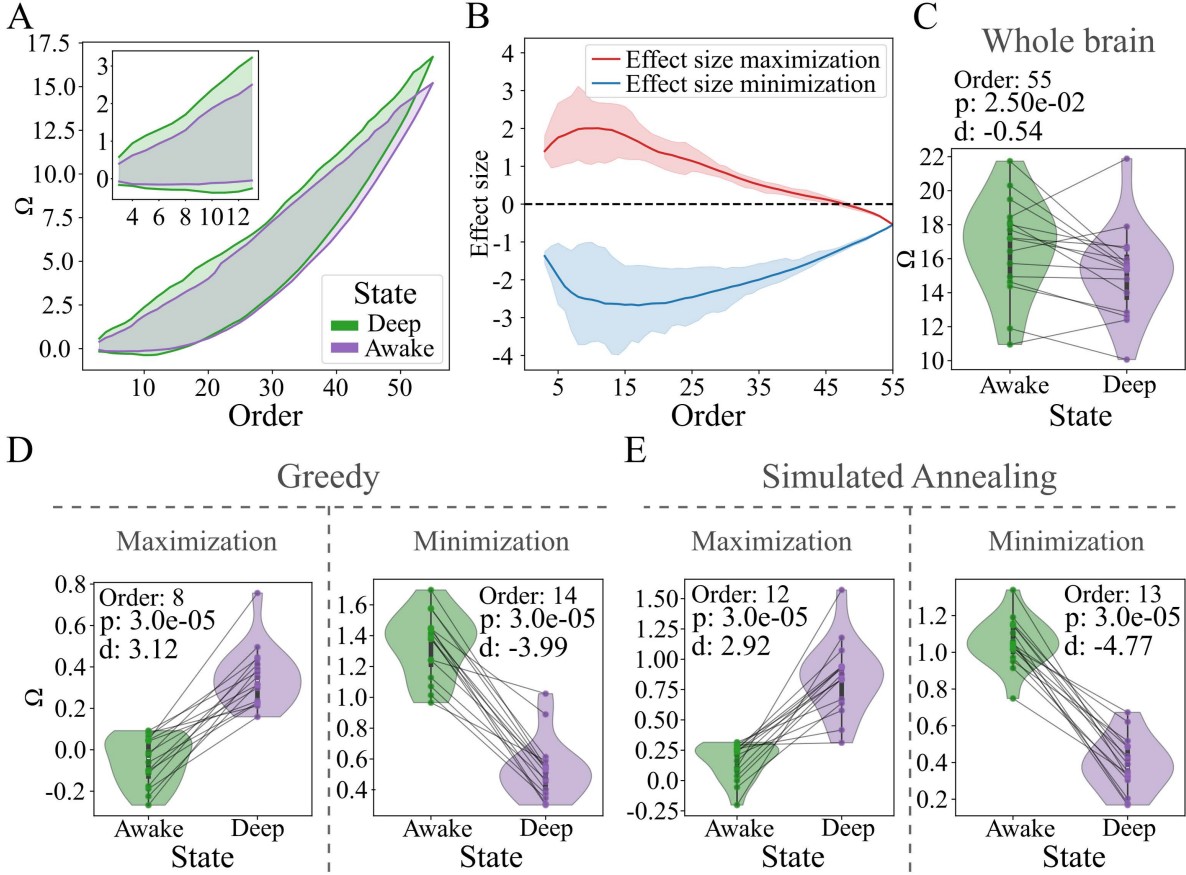

**Fig 5. A) Estimated maximum and minimum $\Omega$ via the GA for awake (green) and deep anesthesia state (purple).** Inset shows the reduction of minimum $\Omega$ at lower orders of interaction. **B)** Average maximum (red) and minimum (blue) effect size obtained from a GA tailored to amplify the difference between the two conditions. Shaded areas denote the range from the minimum to the maximum value for each optimization procedure. **C)** Distribution of $\Omega$ for the whole-brain in awake (green) and deep anesthesia state (purple). Each dot is a subject and lines connect their respective value in both conditions. Despite the trend to reduce redundancy, no significant difference was found (Wilcoxon $p > 0.001$) **D)** Distribution of the $n$-plets that maximizes (left) and minimizes (right) the effect size obtained by the GA. **E)** Same as **D**, but for the $n$-plets obtained with the SA algorithm. Order is the number of elements in $n$-plets, $p$ is the Wilcoxon p-value and $d$ is the Cohen's ***d***.

In summary, these findings suggest that anesthesia compresses the range of $\Omega$ values in the brain, reducing both maxima and minima, and diminishes the dominance of synergy across interactions between distinct networks while reducing redundancy within same-network regions.

## Analysis of large database of synthetic and real-world systems

To demonstrate the potential of THOI and its application to a wide spectrum of complex systems, we analyzed 920 datasets, including both synthetic and real-world data, with system sizes ranging from 5 to 20 [43]. We exhaustively computed $TC$, $DTC$, $\Omega$, and $S$-information for all variable combinations (from 2 to $N$, the system size) across the entire database in under 20 minutes using a laptop with an Intel i7-13800H processor.

For each dataset, we extracted features characterizing the informational structure, including the maximum, minimum, mean (across all orders of interaction) and whole-system values of the four aforementioned metrics, the mean and standard deviation of pairwise mutual information (MI), the order at which $\Omega$ is maximized and minimized (normalized by $N$),

and the proportion of synergy-dominated $n$-plets relative to the total $n$-plets. Although these features were derived from an exhaustive computation of all interactions, all except the proportion of synergy-dominated $n$-plets can be efficiently estimated using GA or SA, making them applicable to larger systems.

We found strong correlations among several features (Fig 6A), particularly those related to the maximum, mean, whole-system, and pairwise mutual information (MI). Using principal component analysis (PCA), we observed that the first principal component (PC) explained over 65% of the variance (Fig 6B), with values primarily reflecting features tied to the maximum, mean, whole-system, and MI (Fig 6C). We termed this component 'Overall Interdependence' as increases in the grouped features indicate stronger overall interdependencies. The negative loadings correspond to synergy-related features (proportion of synergistic n-plets, order of min $\Omega$, and min $\Omega$), which mirror their negative values in the correlation matrix (Fig 6A). This indicates that strong low-order interdependence is associated with increasing redundancy at higher orders, whereas synergy-dominated interactions preferentially arise when low-order dependencies are weaker. The second PC primarily loaded onto features associated with the minimum, leading us to term it 'Overall Independence' as increases in these features reflect departure from independence. Importantly, the orthogonality of the PCs ensures that increases in overall interdependence do not imply decreases in overall independence, as these properties can vary within and across orders of interactions. The third PC primary loaded on features related to the proportion of synergy-dominated $n$-plets and to the order at which the $\Omega$ is maximized and minimized. We termed this PC 'Proportion of Synergistic $n$-plets', as measuring any of these 3 features will capture the proportion of synergy-dominated $n$-plets. This component captures a regime complementary to Overall Interdependence, characterized by weak low-order interactions and a high numerosity of synergy-dominated n-plets. It reflects both the prevalence of synergistic interactions and their depth across orders. Together with the first component, this component highlights that synergy is not driven by strong pairwise dependence, but is instead predominantly realized at higher orders. It is remarkable that even when the proportion of synergistic $n$-plets can't be directly measured in large systems, our results suggest that it can be approximated by the order of the maximum or minimum $\Omega$. Finally, the fourth PC (which in addition to the other three PCs accounts for approximately 95% of the variance), loaded mainly into the maximum, minimum, mean and whole-system $\Omega$-related measures. Accordingly, we termed it 'O-information'.

These results highlight that THOI provides an efficient and ready-to-apply approach for estimating features that characterize HOI in complex systems. Our analysis suggests that these features capture key properties such as the overall interdependence, overall independence, the proportion of synergy-dominated interactions, and the synergy-redundancy balance. Together, these dimensions provide a general framework for understanding the informational structure of complex systems.

## Conclusions

In this article, we introduced THOI, a novel, accessible and efficient Python library designed to compute HOI in complex systems. By leveraging Gaussian copulas and optimized matrix operations in PyTorch, THOI addresses several key challenges in the estimation of HOI, including the combinatorial explosion and the difficulty of directly estimating probability distributions. Our results demonstrate that THOI significantly outperforms existing tools in terms of computational efficiency and capability to run in ordinary laptops. These findings validate THOI's utility and open the door to broader applications in the study of complex systems.

THOI's batch-based architecture integrated with PyTorch enables the parallel processing capabilities of modern CPU, GPU, and TPU architectures. Our comparison of THOI's performance against existing libraries reveals a significant improvement in processing time and memory usage. For example, for a moderate 30-variable system, THOI was able to compute $\Omega$ for all the interactions in a fraction of the time required by alternative open-source libraries, such as those implemented in JIDT, HOI Toolbox and HOI while consuming less than 3Gb of RAM. Therefore, our library offers significant advantages in terms of accessibility, speed and memory efficiency through batch processing. It is designed as well

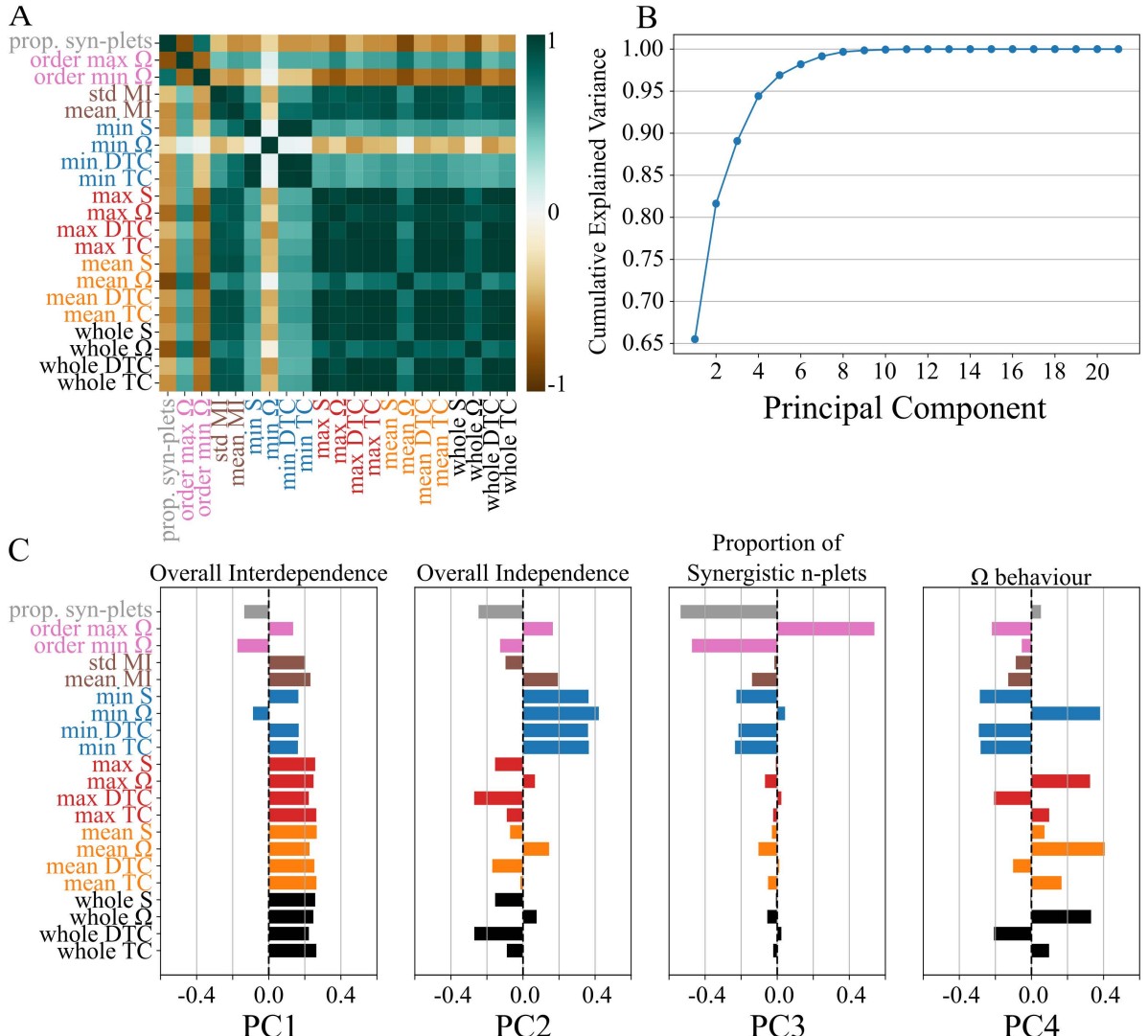

**Fig 6. A) Spearman correlation matrix of features across datasets.** Colors code different types of features. 'prop. syn-plets' is the proportion of synergy-dominated *n*-plets out of the total number of *n*-plets. 'order max $\Omega$' and 'order min $\Omega$' is the order where $\Omega$ was maximized and minimized, respectively, normalized by the system size. 'mean MI' and 'std MI' are the mean and standard deviation of the pairwise mutual information for each dataset. The prefix 'whole' indicates that the whole system was considered (i.e., all the system variables). **B)** Cumulative explained variance associated with each PC after PCA. The first four components capture approximately 95% of the variance. **C)** Values of the first four PCs on each feature. Colors are the same as in **A**. PC1 captures the overall interdependencies, by grouping together all the 'max', 'mean', 'whole' and 'MI' related features. PC2 captures the overall independence by grouping together all the 'min' related features. PC3 captures the proportion of synergy-dominated interaction by grouping together 'prop. syn-plets' and the order at which $\Omega$ is maximized and minimized. PC4 captures the behavior of $\Omega$, by grouping together its maximum, minimum, mean and whole-system values.

to perform population-level and other on-the-fly analyses without storing exponentially scaling results. Furthermore, to provide a more complete description of the informational structure of the data, it computes all four measures (*TC*, *DTC*, $\Omega$ and *S*), while other toolboxes require separate computations for each. This increased efficiency not only makes THOI a practical tool for researchers but also enables a level of scalability that was previously unattainable using traditional approaches.

To validate THOI's performance, we conducted tests on synthetic datasets with known ground-truth values of $\Omega$, specifically PGMs (see Supporting information Probabilistic graphical models (PGM)). Results shown in Fig 4 showed that THOI heuristics where able to capture inherent properties of the PGM. When optimizing for redundancy, heuristics led to a linear increase in $\Omega$ because, in a PGM R-system, information is copied across its parts. Therefore, adding more variables of the R-system increases the amount of redundancy linearly. In contrast, optimizing for synergy resulted in an exponential decrease in $\Omega$. This occurred because in a PGM S-system, synergy is established through a collider variable, which if not considered within the sub selection of variables, no synergy would be detected. The independent sub-system and the S sub-system are included almost simultaneously. The close agreement between THOI's results and the ground-truth values highlights its robustness and reliability as a tool for quantifying HOIs in large complex systems.

Moreover, our validation using empirical data from fMRI studies further confirmed the practical utility of THOI. When applied to fMRI data from human participants, THOI uncovered qualitative and quantitative changes on brain interdependencies induced by deep anesthesia. These findings align with existing literature, which suggests that HOIs in brain networks may be critical for understanding changes in brain dynamics under different states of consciousness [16,17].

Our final demonstration of THOI potential included the exhaustive analysis of all possible interactions on more than 900 datasets in less than 30 minutes in a standard laptop. By analyzing the behavior of key features of the informational structure of HOIs across datasets, we proposed a general framework to characterize complex systems in general. These key features can be easily estimated using the GA and SA and provide complementary information –as revealed by the PCA– about the multi-order level of interdependence, the multi-order level of independence, the proportion and of synergy-dominated interactions and the overall synergy-redundancy balance. These results establish THOI as a superior tool in terms of accessibility, speed, and scalability, significantly lowering barriers to better understanding complex interactions in multivariate systems.

Despite its many advantages, THOI presents several limitations primarily due to its reliance on Gaussian-based methods—the Gaussian estimator of joint entropies and the Gaussian copula approach—which may overlook significant interdependencies present in higher moments or non-Gaussian data distributions. While alternative estimators (e.g. non parametric or kernel based), such as those implemented in JIDT [28] or the HOI toolbox [32], could provide more robust HOI estimations, they may compromise performance and scalability, making it challenging to analyze larger systems. Additionally, THOI is not yet optimized for high-performance computing architectures with multiple GPUs, requiring custom scripts to handle this kind of architecture. However, one of THOI's main focus was to be able to run in standard computational setups to ensure accessibility. Future developments should address these Gaussian-related limitations and enhance computational optimizations to better leverage current and emerging HPC technologies, thereby improving the tool's robustness and scalability without sacrificing accessibility.

In the endeavor to embrace real-world large complex systems, previous studies have already employed heuristic algorithms to optimize $\Omega$. For example, in [46] the authors proposed a SA approach and particle swarm optimization to identify synergy-dominated interactions. THOI introduces a more flexible implementation of SA and a GA capable of searching for optimal $n$-plets for any of the four measures $TC$, $DTC$, $\Omega$, or $S$. This framework also offers flexibility, allowing custom metrics derived from these higher-order interactions (HOI)—such as effect size, classification accuracy, or regression performance—to be optimized across datasets. This adaptability is particularly valuable in scenarios where the measure of HOI that is mathematically optimal may not align with the most effective solution for a specific practical application, as discussed in [50].

Despite these advancements, these heuristics remain general-purpose optimizers that do not fully exploit the underlying mathematical properties of the $TC, DTC$, $\Omega$, and $S$ measures. Future research could focus on developing more principled, measure-specific optimization strategies. For instance, leveraging the recently introduced gradients of $\Omega$ [25] within a gradient-based optimization framework may significantly refine the search process and yield even more robust results.

The development of THOI has important implications for the study of complex systems across multiple domains. In neuroscience, where HOI are thought to play a crucial role in cognition and consciousness [9,21,45,51–53], THOI offers a

powerful tool for uncovering the collective behavior of neural systems, that coupled with mechanistic whole brain models [54], could provide novel insights about the biophysical origin of observed high-order interdependencies [55]. In macro-economics, where interconnectedness between financial entities and the non-linear dynamics of the economy are key to understanding crises and systemic risks [6,7], THOI can aid in identifying emergent patterns of behavior that may not be immediately apparent through conventional analysis. Similarly, in the study of the brain [56–58], ecological networks [59,60], social systems [61,62], and even musical analysis [27].

## Code availability

The Python library presented in this study, `THOI`, is open-source and publicly accessible under the MIT license. It can be found at:

- GitHub repository: https://github.com/Laouen/THOI/releases/tag/v0.2.33

- Python Package Index (PyPI): https://pypi.org/project/thoi/0.2.33/

- Archived release on Zenodo: https://zenodo.org/records/15020522

All code necessary to reproduce the analyses demonstrated in this paper is available in the accompanying GitHub repository at https://github.com/Laouen/thoi_tutorials. Additionally, a stable version of the code has been archived on Zenodo for reproducibility (https://zenodo.org/records/15020335). The provided code consists primarily of Python scripts, with some supplementary Java scripts to get JIDT times and Jupyter Notebooks for the figures. Detailed README files and a `requirements.txt` file are included to facilitate straightforward environment setup and step-by-step reproduction of all analyses.

## Supporting information

**S1 Text. Mathematical basis.**
(PDF)

## Author contributions

**Conceptualization:** Laouen Belloli, Rubén Herzog.

**Formal analysis:** Laouen Belloli, Rubén Herzog.

**Methodology:** Laouen Belloli, Pedro A. M. Mediano, Rodrigo Cofré, Rubén Herzog.

**Software:** Laouen Belloli.

**Supervision:** Pedro A. M. Mediano, Rodrigo Cofré, Diego Fernandez Slezak, Rubén Herzog.

**Validation:** Pedro A. M. Mediano, Rodrigo Cofré.

**Writing – original draft:** Laouen Belloli, Rubén Herzog.

**Writing – review & editing:** Laouen Belloli, Rubén Herzog.

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
