## [Decision Letter · Decision Letter 0]

5 Jan 2026

PONE-D-25-41547THOI: An efficient and accessible library for computing higher-order interactions enhanced by batch-processingPLOS One

Dear Dr. BELLOLI,

Thank you for submitting your manuscript to PLOS ONE. After careful consideration, we feel that it has merit but does not fully meet PLOS ONE’s publication criteria as it currently stands. Therefore, we invite you to submit a revised version of the manuscript that addresses the points raised during the review process.

If applicable, we recommend that you deposit your laboratory protocols in protocols.io to enhance the reproducibility of your results. Protocols.io assigns your protocol its own identifier (DOI) so that it can be cited independently in the future. For instructions see: https://journals.plos.org/plosone/s/submission-guidelines#loc-laboratory-protocols. Additionally, PLOS ONE offers an option for publishing peer-reviewed Lab Protocol articles, which describe protocols hosted on protocols.io. Read more information on sharing protocols at . Additionally, PLOS ONE offers an option for publishing peer-reviewed Lab Protocol articles, which describe protocols hosted on protocols.io. Read more information on sharing protocols at https://plos.org/protocols?utm_medium=editorial-email&utm_source=authorletters&utm_campaign=protocols..

We look forward to receiving your revised manuscript.

Kind regards,

Javed Iqbal, PhD

Academic Editor

PLOS One

**Journal Requirements:**

2. Please update your submission to use the PLOS LaTeX template. The template and more information on our requirements for LaTeX submissions can be found at http://journals.plos.org/plosone/s/latex..

3. We note that your Data Availability Statement is currently as follows:

“All relevant data are within the manuscript and its Supporting Information files.”

Reviewers' comments:

Reviewer's Responses to Questions

**Comments to the Author**

1. Is the manuscript technically sound, and do the data support the conclusions?

Reviewer #1: Yes

Reviewer #2: Yes

2. Has the statistical analysis been performed appropriately and rigorously? 

Reviewer #1: Yes

Reviewer #2: Yes

3. Have the authors made all data underlying the findings in their manuscript fully available?

Reviewer #1: Yes

Reviewer #2: Yes

4. Is the manuscript presented in an intelligible fashion and written in standard English?

Reviewer #1: Yes

Reviewer #2: Yes

5. Review Comments to the Author

Reviewer #1: I found the present manuscript quite interesting, since it provides a very powerful tool to compute high-order interactions measures in complex systems, including the brain. The structure of paper is appropriate and figures are of great quality. Improvement in the performance of the proposed python library compared with other toolboxes in the literature is also very high. Therefore I recommend publication of the present manuscript

Reviewer #2: In this paper, Belloli et al. introduce a new toolbox for the efficient computation of higher-order information-theoretic measures. They not only evaluate its performance on simulated data, but also test it on multiple datasets, including human fMRI. This work represents a valuable contribution to the field, and I am highly supportive of its publication. I only have a few minor comments and questions that the authors may wish to address to improve the clarity of the narrative and the robustness of the paper.

1. Figure 2 (panels C–F) could be clearer. It took me some time to understand what these panels show, and I am still not entirely certain I interpreted them correctly. Does each line correspond to one of the 20 variables in the ground truth? For example, in panel E, does an interaction order of 40 mean that all blue variables contribute to minimizing \omega, while at an interaction order of 80 the variables shown in light red also contribute?

2. One aspect the authors could consider testing is the stability of these algorithms. If the algorithms were run multiple times, would they consistently identify the same subset that minimizes/maximizes \omega for a given interaction order?

3. In the analysis of the fMRI dataset, the authors could consider adding two panels to Figure 3 to show on the cortical surface which n-plets (brain regions) drive the statistically significant increases and decreases in \omega.

4. In the analysis of large databases of synthetic and real-world systems, the authors report correlations between features of the information-theoretic measures. How do the authors interpret the finding that several features (e.g., pro syn n-plets, order min \omega, and min \omega) are strongly anticorrelated across datasets?

6. PLOS authors have the option to publish the peer review history of their article (what does this mean?). If published, this will include your full peer review and any attached files.). If published, this will include your full peer review and any attached files.

.

Reviewer #1: No

Reviewer #2: No

---

## [Author Response · Author response to Decision Letter 1]

24 Mar 2026

Please see the uploaded document titled “Response to Reviewers” for our detailed point-by-point responses and summary of revisions.

---

## [Decision Letter · Decision Letter 1]

9 Apr 2026

THOI: An efficient and accessible library for computing higher-order interactions enhanced by batch-processing

PONE-D-25-41547R1

Dear Author,

We’re pleased to inform you that your manuscript has been judged scientifically suitable for publication and will be formally accepted for publication once it meets all outstanding technical requirements.

An invoice will be generated when your article is formally accepted. Please note, if your institution has a publishing partnership with PLOS and your article meets the relevant criteria, all or part of your publication costs will be covered. Please make sure your user information is up-to-date by logging into Editorial Manager at Editorial Manager® and clicking the ‘Update My Information' link at the top of the page. For questions related to billing, please contact  and clicking the ‘Update My Information' link at the top of the page. For questions related to billing, please contact billing support..

Kind regards,

Javed Iqbal, PhD

Academic Editor

PLOS One

Additional Editor Comments (optional):

Reviewers' comments:

Reviewer's Responses to Questions

**Comments to the Author**

1. If the authors have adequately addressed your comments raised in a previous round of review and you feel that this manuscript is now acceptable for publication, you may indicate that here to bypass the “Comments to the Author” section, enter your conflict of interest statement in the “Confidential to Editor” section, and submit your "Accept" recommendation.

Reviewer #2: All comments have been addressed

2. Is the manuscript technically sound, and do the data support the conclusions?

Reviewer #2: Yes

3. Has the statistical analysis been performed appropriately and rigorously? 

Reviewer #2: Yes

4. Have the authors made all data underlying the findings in their manuscript fully available?

Reviewer #2: Yes

5. Is the manuscript presented in an intelligible fashion and written in standard English?

Reviewer #2: Yes

6. Review Comments to the Author

Reviewer #2: The authors carefully addressed all my comments, and I'm very happy to recommend this paper for publication.

7. PLOS authors have the option to publish the peer review history of their article (what does this mean?). If published, this will include your full peer review and any attached files.). If published, this will include your full peer review and any attached files.

.

Reviewer #2: No

---

## [Editor Report · Acceptance letter]

PONE-D-25-41547R1

PLOS One

Dear Dr. BELLOLI,

I'm pleased to inform you that your manuscript has been deemed suitable for publication in PLOS One. Congratulations! Your manuscript is now being handed over to our production team.

Kind regards,

on behalf of

Dr. Javed Iqbal

Academic Editor

PLOS One